# Influence of Plastic Deformation on Microstructural Evolution of 100Cr6 Bearing Ring in Hot Ring Rolling

**DOI:** 10.3390/ma13194355

**Published:** 2020-09-30

**Authors:** Guanghua Zhou, Wenting Wei, Qinglong Liu

**Affiliations:** 1School of Materials Science and Engineering, Wuhan University of Technology, Wuhan 430070, China; zhoughjixi@163.com; 2Hubei Key Laboratory of Advanced Technology for Automotive Components, Wuhan University of Technology, Wuhan 430070, China; liuql_whut@126.com; 3Hubei Engineering Research Center for Green Precision Material Forming, Wuhan 430070, China; 4School of Automotive Engineering, Wuhan University of Technology, Wuhan 430070, China; 5Hubei Research Center for New Energy & Intelligent Connected Vehicle, Wuhan University of Technology, Wuhan 430070, China; 6Hubei Collaborative Innovation Center for Automotive Components Technology, Wuhan University of Technology, Wuhan 430070, China

**Keywords:** 100Cr6, hot ring rolling, microstructure, EBSD, Vickers hardness

## Abstract

The hot ring rolling technology as the crucial procedure for the manufacture of bearing rings plays an important role in determining the final microstructure of bearing rings. In this work, the influence of the hot ring rolling process on the microstructural evolution of 100Cr6 bearing rings was investigated using a three-dimensional (3D) numerical model and microstructural characterization. It was found that the significant microstructural refinement occurs at the different regions of the rings. However, owing to the non-uniform plastic deformation of hot rolling, the refinement rate of grain size and decrease of pearlite lamellar spacing (PLS) also showed uniformity at different regions of the rings. Furthermore, the degree of grain refinement had been limited with the increase of rolling reduction. Due to the refined grain size and decreased PLS, the Vickers hardness increased with the increase of rolling reduction. Moreover, the Vickers hardness from the outer surface to the inner surface of the ring is asymmetrical u-shaped, which had the law of lower hardness in the center area and higher hardness on the surface.

## 1. Introduction

Hot ring rolling is an advanced incremental metal-forming process. The rings manufactured by hot ring rolling not only have the characteristics of high geometric accuracy, energy and raw material saving, but also have outstanding microstructure performance [1,2]. Due to the wide application of rotary parts such as high-speed railway and wind power bearing, hot ring rolling technology has gradually attracted more and more attention. As a new method of plastic deformation, a large number of scholars have contributed many valuable works in design, theoretical calculation and simulation. Hua et al. [3,4] analyzed the extremum parameters and ring stiffness condition in ring rolling. The deformation behaviors and blank design of profile ring rolling were also thoroughly analyzed [5,6,7]. Yang et al. [8] presented the effects of blank size on the uniformity of strain and temperature distribution during hot rolling. The design of ring rolling for a large-scale ring containing blank dimension and process variables was also put forward [9]. Zhu et al. [10] put forward four blank design principles for conical-section ring rolling and the dimension effects of blanks on dimensional accuracy and thermo-mechanical parameters distribution were explored.

100Cr6 bearing steel plays an irreplaceable role in bearing rings and rolling elements. Many scholars have made their dedication and continued efforts in the experimental research of 100Cr6. With the establishment of macro-microscopic constitutive equations of 100Cr6 steel [11,12], Gu et al. [13] investigated the multi-field coupled numerical simulation of microstructure during hot rolling. The distribution and evolution of microstructure characteristics was presented. Deng et al. [14] discovered that the grain can be refined effectively by increasing the degree of rolling deformation. Based on the hot radial ring rolling of 100Cr6 steel, Guo et al. [15] found that grain refinement had been limited at a certain sufficient plastic deformation, and further deformation will not help in overcoming these limits. The significant influence of hot forming parameters on grain size has been explored [16]. Recently, Ryttverg et al. [17,18] investigated the cold ring rolling process of 100Cr6 steel using electron backscatter diffraction (EBSD) and scanning electron microscope (SEM). The unique phenomenon that the microstructure heterogeneity and texture change during rolling in the radial direction of the ring was proposed. When plastic deformation is carried out on the same kind of material, the metal flow behavior of the sample at high temperature and room temperature should be similar, but the microstructural evolution rule should show different characteristics in consideration of the influence of temperature on microstructure. However, the studies of microstructural evolution after hot rolling mainly explored the constitutive equation [11,12,19,20]. In the meantime, for 100Cr6 bearing steel during hot ring rolling forming, most of the research mainly focuses on the metal flow law and control method [21,22,23]. Therefore, the effect of hot ring rolling on the microstructure and mechanical properties needs systematic investigation.

In this work, to analyze the evolution and distribution laws of grain size, pearlite lamellar spacing (PLS) and Vickers hardness, a series of experiments of the hot rolling ring with ball groove are conducted by performing EBSD and SEM technologies. Moreover, a three-dimensional (3D) finite element (FE) model of hot ring rolling is established to illustrate the evolution of effective plastic strain and the effect on the microstructure. According to the analysis of the experimental and simulative results, the microstructure evolution of hot ring rolling is revealed.

## 2. Hot Ring Rolling and Experimental Tests

### 2.1. Finite Element Simulation

A valid 3D finite element model is established based on simufact finite element analysis software (16.0, MSC software company, Hamburg, Germany). The precise material characteristic is necessary to obtain the more reliable and accurate simulation results. Hence, the constitutive modeling for flow behavior [11] and microstructural evolution [12] is applied. The schematic illustration and dimension parameter of the hot ring rolling process are demonstrated in Figure 1. The detailed dimension parameters are presented in Table 1, which is consistent with the relevant parameters of the experiment. The geometrical shape and effective plastic strain evolution in simulation of the rolled rings under various rolling time is shown in Figure 2. The effective plastic strain is computed by incrementally integrating the local straining rates [24].

### 2.2. Material and Specimen Preparation

The chemical composition of 100Cr6 bearing steel used in this study mainly consists of 0.98% C, 0.20% Si, 0.31% Mn, 1.42% Cr and balanced Fe. The specimens were received from a forged state bar with a diameter of 110 mm. The specimens were first heated to nearly 1050 °C and then moved to the next procedures which contained upsetting, punching and leveling. After the initial blank making process, the surface temperature of the blank was about 920 °C. Then, the ring rolling was produced and obtained deformed rings with different rolling reductions. After the ring rolling was finished, the surface temperature of the rings was between 840 and 870 °C in different rolling reduction. Afterwards, the rolled rings were placed in a draught fan for rapid cooling to avoid the appearance of reticular carbides.

The rolling reductions of 45.18%, 53.81% and 70.05% for rolled ring 1, rolled ring 2 and rolled ring 3 were applied. The radial diameter increased rapidly while the axial height had no significant change with the increase of rolling reduction. The rolling reduction, which means the thickness reduction rate of the rings, is given as [25]:(1)$$\mathrm{Rolling}\;\mathrm{reduction}\;=\;\frac{\left(D_{0}-d_{0})-(D-d_{2}\right)}{\left(D_{0}-d_{0}\right)}$$
where $D_{0}$ and $d_{0}$ are the outer and inner diameter of the blank, respectively.

The blank and rolled rings are presented in Figure 3 and the corresponding dimensions are listed in Table 2. The mean grain size of the raw material was 48.3 μm, as shown in Figure 4, and the Vickers hardness was approximately 390 HV.

### 2.3. Microstructure and Mechanical Property Test

To analyze the microstructural evolution of the rolled rings under different rolling reductions, the samples A, B and C (shown in Figure 5), which represented the outer surface, center area and inner surface of the rolled rings respectively, were selected, and the dimension of the test sample is 3 × 6 mm. For precisely quantifying the grain size and PLS, five fields were taken from each sample.

Before further tests of microstructure and mechanical property, the sample preparation process was strictly controlled. The samples were mechanical grinded with abrasive paper from 150 to 2000 level and polished with diamond suspension with particle size of 5, 2.5 and 1 μm for 5 min, respectively. In the end, vibration polishing with colloidal silica suspension was used for more than 12 h to eliminate the stress of the surface. The test area was 240 μm × 200 μm, which contains about 100–150 grains. The used step size was 0.4 μm and the percentage of indexing was more than 90%. Then, the mean grain size was measured by Image pro plus analysis software (Version 6.0.0.260, Media Cybernetics Inc., Singapore).

The presentation of grain size was carried out by the method of electron backscatter diffraction (EBSD). The EBSD pattern was acquired by a detector that was attached to the field emission scanning electron microscopy (FESEM) system (JEM-7500F) (Japan Electron Optics Laboratory, Beijing, China). For observing the pearlite lamellar, the samples were examined by the SEM using a JSM-IT300 microscope (Japan Electron Optics Laboratory, Beijing, China). Vickers hardness was tested by a HV-1000 hardness testing machine (Laizhou Huayin Test Instrument CO, LTD, Laizhou, China) with 200 g load and dwell time of 5 s. The test region ranged from the outer surface to the inner surface and the distance between two adjacent points was 1 mm.

## 3. Results and Discussion

### 3.1. The Evolution of Grain Size Refinement

Figure 6 shows the EBSD maps of the selected samples during hot ring rolling. In the figure, disparate color regions mean different grains. The different color of adjacent grains indicates that there is a certain misorientation. The larger obvious difference of the color, the larger the misorientation angle between the grains. After hot ring rolling, the grain size is significantly reduced compared with the initial state (Figure 4a), and the grain size decreases obviously with the increase of rolling reduction. The grain size comes up to the finest at the largest deformation of 70.05%. Besides, there is big discrepancy between samples A, B and C. The grain size of sample B is significantly larger than other positions, and the grain size of the sample A and C at deformation of 53.81% are comparable with that of the sample B at deformation of 70.05%. That means the local grain size is refined as the rolling reduction increases. Nevertheless, the degree of refinement is variable in different position of the rings.

The statistical distribution of grain size of sample A is listed in Figure 7a. There is a similar distribution for different samples. When the rolling reduction is small, there are more large size grains, and the mean grain size is 26.9 μm. With the increase of rolling reduction, the amount of small size grains increases gradually. The mean grain size is 13.9 μm when the rolling reduction is 70.05%. For sample B, the mean grain size is 33.9 μm in the deformation of 45.18%, as shown in Figure 7b. When the rolling reduction is 53.81%, the main distribution of grain size is between 10 and 30 μm, and the mean grain size is 20.9 μm. When the deformation is further increased to 70.05%, the grains are further refined, and the grain size reduces to 15.7 μm. The effect of plastic deformation on grains is more significant for sample C. The grain refinement phenomenon is relatively obvious. The distribution of grain size is presented in Figure 7c, and the mean grain size is 15.5, 15.1 and 13.3 μm with the increase of rolling reduction. It can definitely be found that the grain size at different positions have the same evolution law.

The evolution law of mean grain size is presented in Figure 7d. The mean grain size in the inner surface of the rolled rings is smaller than that of the other positions, and the mean grain size is decreased with the increase of rolling reduction at each sample. Moreover, the rate of grain refinement is most serious in the inner zone, second in the outer zone and lowest in the center zone. While, there is no difference in grain size at different positions within the error range when the rolling reduction reaches 70.05%, because the grain size does not seem to decrease indefinitely. The grain size becomes stable when the rolling reduction reaches a certain level, which is a similar conclusion as in Reference [15].

Figure 8 shows the effective plastic strain of the selected samples during the hot ring rolling process. The effective plastic strain is computed by incrementally integrating the local strain rates, which are already integrated into the simufact software. The effective plastic strain of sample C remained largest and that of the sample B remained smallest during the whole hot ring rolling process. The rapid increase of local strain at sample C leads to the increase of deformation storage energy compared with other samples. With the deformation storage energy improving, the dislocation density is improved and provides more driving force for dynamic recrystallization and migration of grain boundaries [26]. It offers a prerequisite for rapid decreasing of the grain size at sample C. The grain size of sample C is the smallest, which can be seen in Figure 7d. The grain size of sample B decreases relatively slowly. With the increase of rolling reduction, the grain size of samples A and B decrease gradually, while that of sample C has no significant decline because the grain refinement limit has been reached [15].

The pole figures analysis of the blank shows rather strong orientation densities, while the maximum value of orientation densities decreases gradually with the increase of rolling reduction. When the rolling reduction is 70.05%, the distribution of orientation densities is more uniform, as seen in Figure 9. In the hot ring rolling process, two variations of microstructural evolution exist. Plastic deformation was dominated by dislocation movement and dynamic recrystallization was dominated by recovery, and nucleation occurred alternately. Plastic deformation forms a deformation texture and dynamic recrystallization results in a recrystallization texture. The two processes are carried out alternately at the same time, neither texture can be fully developed. This results in a weak texture after hot deformation. The homogeneous distribution of texture orientation and the decrease of grain size may further improve the property of the ring after hot ring rolling and heat treatment.

The EBSD image and misorientation profiles of the samples are shown in Figure 10. The misorientation profiles show the distribution of detailed misorientation angles in the interior of the grain. The “point-to-origin” misorientation profiles increase in the grain with the increase of rolling reduction. The sub-grain boundaries are formed within the grains which shows in the “point-to-point” profile [27]. The appearance and increase of the low-angle sub-grain boundary can be clearly found in Figure 10b, and the misorientation profile shows that the angle in “point-to- point” already reaches up to 15°. It means that the formation of high-angle grain boundaries occurs when the misorientation angle exceeds 15°. The increase in rolling reduction leads to the increase of deformation storage energy, which clearly contributes to the formation of new grain boundaries. The accumulation of dislocations and dynamic recrystallization play an important role. The phenomenon of newly formed grain can be observed in Figure 10d. The large grains of the same misorientation are separated into numerous small grains and sub-grains.

The frequency of grain boundaries with different misorientation scales at sample C are shown in Figure 11 and Table 3. The quantity of the misorientation angle lower than 5° is a little higher than that of 15° of the blank. When the rolling reduction reaches to 45.18%, the frequency of the misorientation angle (lower than 10° and 5°) increases obviously, which implies the formation of sub-grain boundaries. With increasing the rolling reduction to 70.05%, the quantity of the misorientation angle lower than 5° decreases, while those higher than 15° increase because of dislocation motion. The main reason for grain refinement is dynamic recovery and recrystallization because of dislocation gliding and climbing [28,29].

### 3.2. The Characteristic of Pearlite Lamellar Spacing

Figure 12 shows the SEM images of the selected samples during hot ring rolling. The PLS significantly reduces compared with that of the blank (300.7 nm), and the PLS decreases obviously with the increase of rolling reduction. The PLS is finest at the largest deformation of 70.05%, and the PLS of sample B is significantly larger than other samples, which has the same evolution law with the grain size. The mean values of PLS are presented in Figure 13a, which can be quantitatively evaluated with the evolution law of the PLS. When rolling reduction reaches to the maximum, the PLS of the selected samples has little variation.

To further explain the interaction of the grain size and PLS, the relationship between them is presented in Figure 13b. In addition to the same variation pattern, the PLS decreases with the decrease of grain size. The finer lamellar spacing can be acquired from the finer-grained sample, which has the same conclusion as in Reference [30].

The cooling rate, deformation rate, rolling reduction and deformation temperature can affect the transformation of the pearlite and the PLS. In this experiment, all conditions are the same except the rolling reduction. As the typical diffusion transformation, the pearlite transformation includes nucleation and nucleus growth. Because the grain size gradually decreases, the number of grain boundaries increases and the dislocation density increases during the hot ring rolling, and the nucleation position increases. The size of pearlite colonies and PLS are significantly reduced at hot deformation due to the dynamic recovery and recrystallization.

### 3.3. Distribution of Vickers Hardness

The distribution from the outer surface to the inner surface of Vickers hardness for rolled rings is presented in Figure 14. The hardness of the inner surface of the blank is about 390 HV and it is about 400 HV when the rolling reduction reaches 45.18%. The hardness continuously rises with the increase of the rolling reduction, and rises up to more than 480 HV when the rolling reduction is 70.05%. It can be clearly seen that the hardness is different along the radial direction. The values of the outer surface are apparently about 20 HV higher than those of the center area, while the inner surface is about 30 HV higher than that of center area. For the rolled ring 3, the hardness at the outer surface is about 465 HV, then reduces to about 440 HV when the distance from the outer surface is 3 mm and increases rapidly up to about 490 HV at the inner surface. The increase of Vickers hardness that occurs with the increase of rolling reduction can be explained by the decrease of grain size and PLS.

The Hall–Petch relation is usually used to illustrate the effect of grain size on yield stress and is given as [31]:(2)$$\sigma_{s}\;=\;\sigma_{0}+k_{HP}d^{-1/2}$$
where $\sigma_{s}$ is the yield stress, $\sigma_{0}$ is a friction stress, d is the grain size and $k_{HP}$ is a constant.

Besides, previous experimental results show that there is a linear positive correlation between the yield strength and Vickers hardness [32,33]. An intuitive interpretation of this relation indicates that the finer grain sizes should lead to higher Vickers hardness. In theory, it is suggested that the refinement of grain size often leads to more grain and sub-grain boundaries. Therefore, more grains need to be coordinated during plastic deformation, leading to higher yield strength and Vickers hardness.

Due to the fact that the grain size of the surface is larger than that of the center area, the Vickers hardness is a little higher in the surface, and the Vickers hardness increases gradually with the decrease of grain size. Although the effect of measurement errors is taken into account, the asymmetrical u-shaped distribution of Vickers hardness can be clearly observed. The linear relationship between grain size and Vickers hardness in logarithmic coordinates of the samples can be found and is presented in Figure 15a.

The relationship between ferrite plate thickness, $L_{3}$, and flow stress, σ, is described as [34]:(3)$$\sigma \;=\;K{\left(L_{3}\right)}^{-1}+X$$
where K is constant and X is the unified representation of the strength of pure annealed iron, solid solution and the distance between the carbide particles.

The ferrite plate thickness and flow stress are inversely proportional, therefore, the decrease of PLS also has an active influence on the increase of Vickers hardness. Figure 15b shows the relationship between PLS and Vickers hardness of the samples based on experimental data. The Vickers hardness in logarithmic coordinates and PLS also revealed a linear trend. The properties of lamellar pearlite depend on the lamellar spacing. The smaller the lamellar spacing is, the higher the strength and hardness of pearlite are. Small lamellar spacing can promote the increase of phase interface and improve the hardness. With the increase of rolling reduction, under the joint action of grain refinement and the decrease of PLS, the Vickers hardness shows the increasing trend and u-shaped distribution from the outer to inner surface.

From the analysis of the experimental data, the slope of ln (Vickers hardness) and grain size is −0.535 × 10^−2^ and that of ln (Vickers hardness) and PLS is −9.156 × 10^−4^. It indicates that the Vickers hardness increases with the decrease of grain size and PLS. After deformation, with the increase of the quantities of defects and severe distortion of the lattice parameter caused by the decrease of grain size and PLS in the material, the hindrance of dislocation is greater, and the hardness is also increased correspondingly. The grain size and PLS can be used to better predict the evolution of hardness and the appropriate hardness can be achieved by adjusting the microstructure.

## 4. Conclusions

In this paper, the effects of rolling reduction on the grain refinement, PLS and Vickers hardness of 100Cr6 bearing steel were investigated. The main conclusions can be summarized as follows:The mean grain size decreased with the increase of rolling reduction. Although the rate of grain refinement at different region of the rings was different, the grain size was almost the same when the rolling reduction reached the maximum in this study. The grain refinement has been limited in 100Cr6 bearing steel during hot ring rolling. Plastic deformation changes the distribution of low- and high-angle grain boundaries and promotes the merging of lower angle to higher angle boundaries via dislocation motion by providing strong dynamic recrystallization driving force.The PLS and grain size have the same variation rule, and the PLS decreased with the increase of rolling reduction. Smaller grains and more grain and sub-grain boundaries provide more driving force and nucleation points for pearlite formation. The finer PLS can be acquired from the finer-grained samples.The thinner PLS and finer grain size can help in improving the Vickers hardness. Due to the non-uniform plastic deformation at different regions in the rings, the Vickers hardness distribution was asymmetrical u-shaped, and increased with the increase of rolling reduction.

## Figures and Tables

**Figure 1 materials-13-04355-f001:**
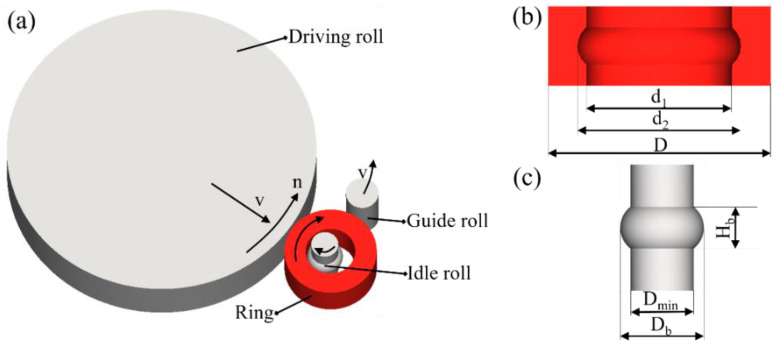
Schematic illustration and dimension parameter: (**a**) hot ring rolling process, (**b**) rolled ring, (**c**) idle roll.

**Figure 2 materials-13-04355-f002:**
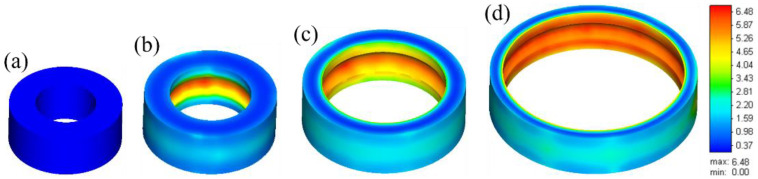
Geometrical shape and effective plastic strain evolution in simulation of the rolled rings under various rolling times (t): (**a**) t = 0 s, (**b**) t = 3.2 s, (**c**) t = 6.5 s, (**d**) t = 10.5 s.

**Figure 3 materials-13-04355-f003:**
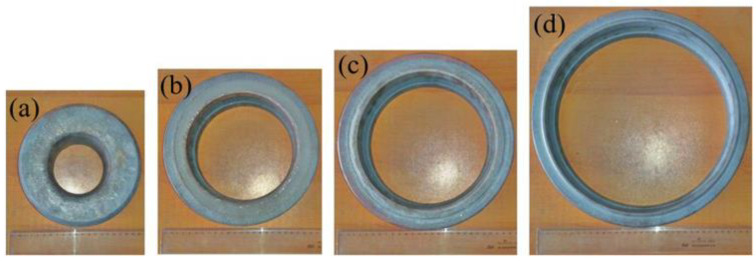
Rings with different rolling reductions: (**a**) blank, (**b**) rolled ring 1, (**c**) rolled ring 2, (**d**) rolled ring 3.

**Figure 4 materials-13-04355-f004:**
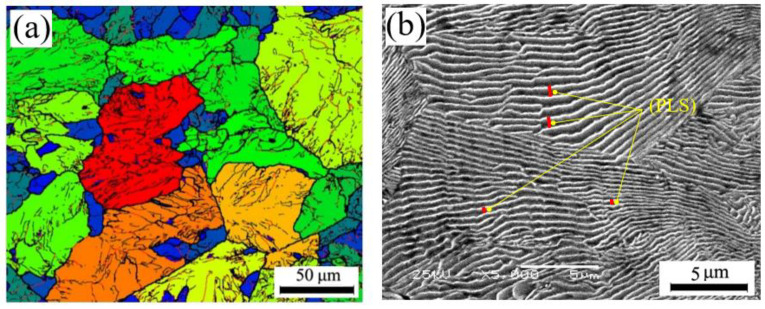
Initial microstructure of the blank: (**a**) EBSD image, (**b**) SEM image.

**Figure 5 materials-13-04355-f005:**
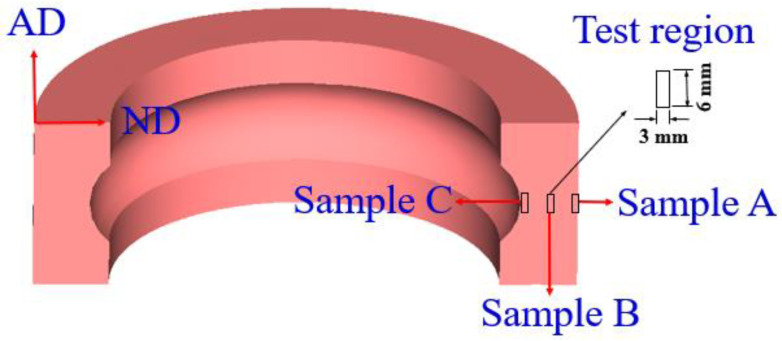
The selected samples and test region of the ring.

**Figure 6 materials-13-04355-f006:**
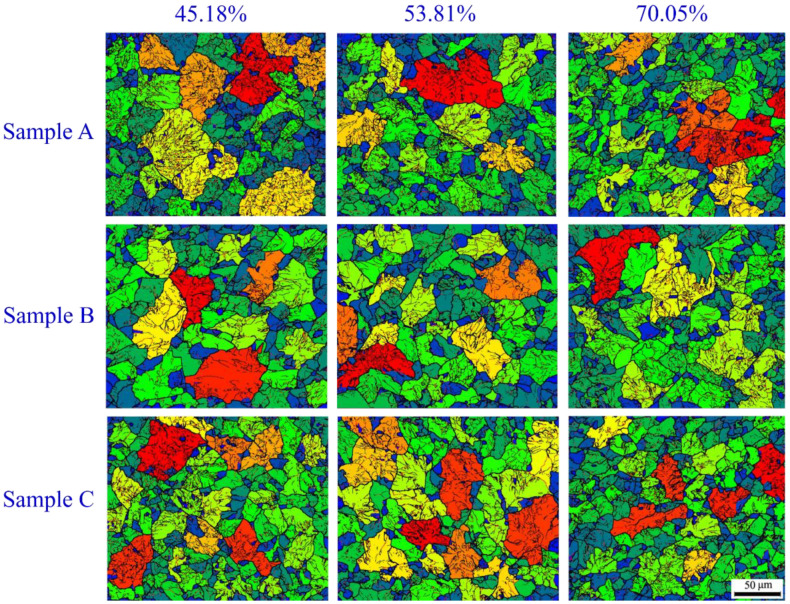
EBSD images of the selected samples.

**Figure 7 materials-13-04355-f007:**
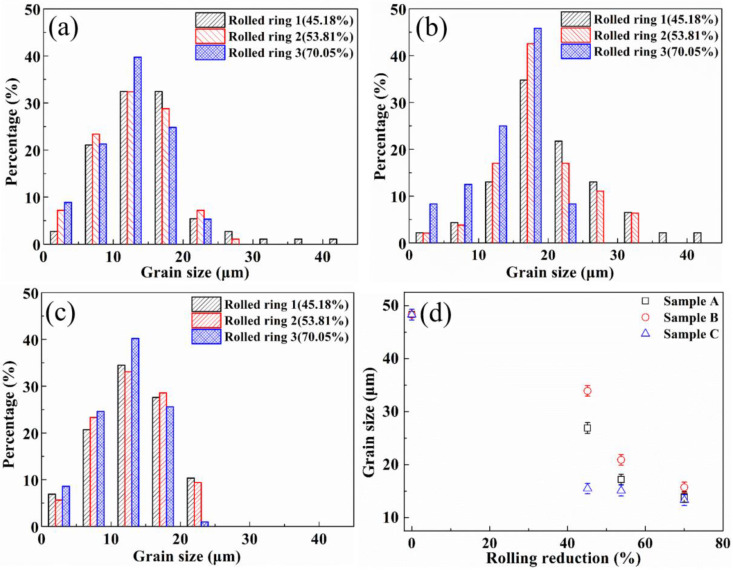
The distribution of the grain size for various samples under different rolling reductions: (**a**) sample A, (**b**) sample B, (**c**) sample C, (**d**) mean size of the grains.

**Figure 8 materials-13-04355-f008:**
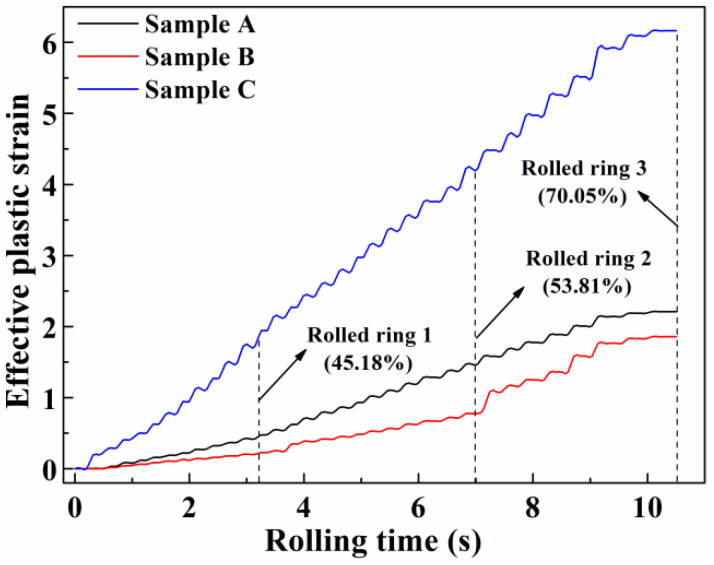
The effective plastic strain of the selected samples.

**Figure 9 materials-13-04355-f009:**
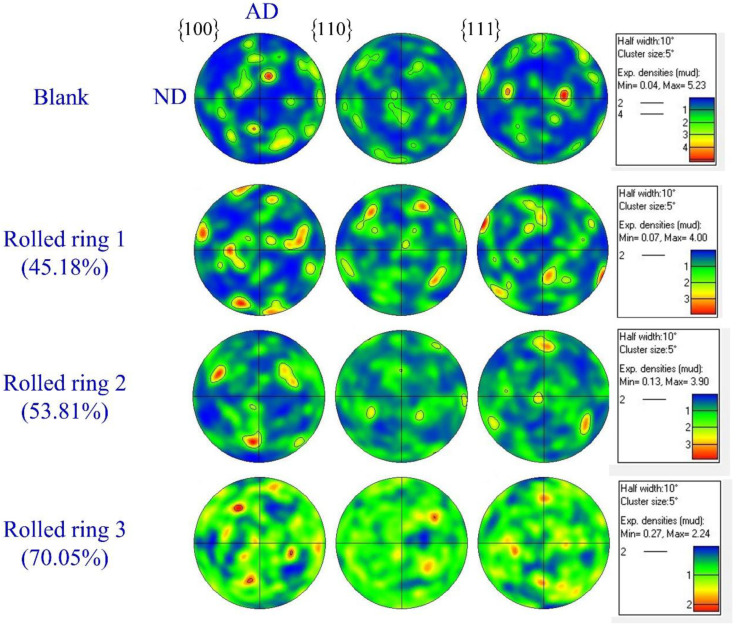
{100}, {110}, and {111} pole figures of the rolled rings at sample C.

**Figure 10 materials-13-04355-f010:**
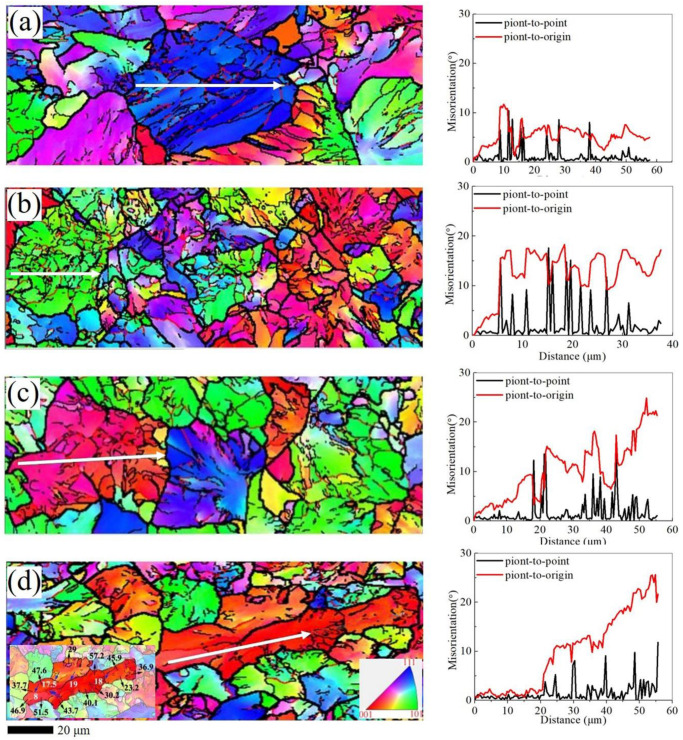
The EBSD image and misorientation profiles during rolling reduction at sample C: (**a**) blank, (**b**) 45.18%, (**c**) 53.81%, (**d**) 70.05%.

**Figure 11 materials-13-04355-f011:**
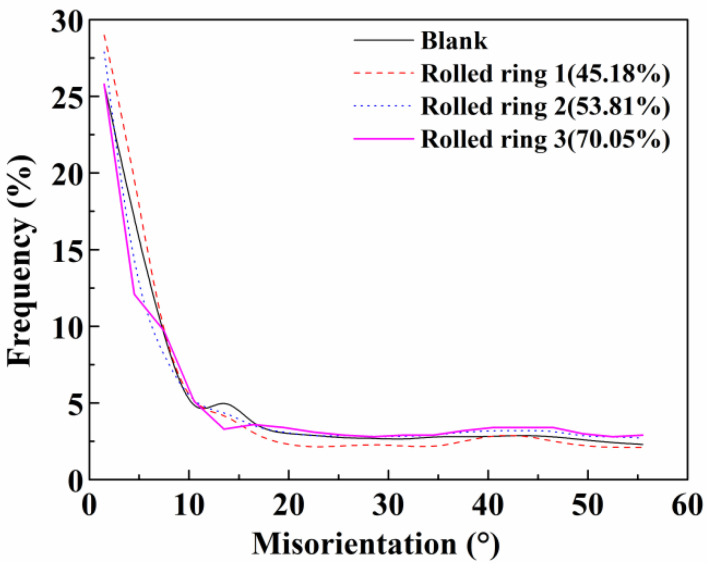
The frequency evolution of misorientation angles at sample C.

**Figure 12 materials-13-04355-f012:**
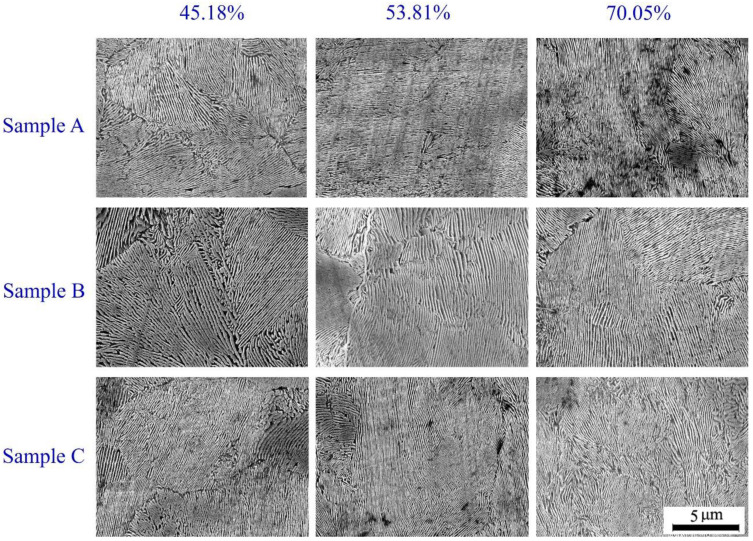
SEM images of the selected samples.

**Figure 13 materials-13-04355-f013:**
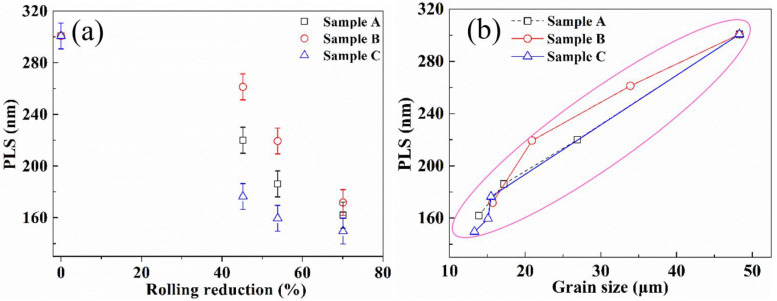
The characteristic of the pearlite lamellar spacing (PLS). (**a**) The mean PLS of the selected samples, (**b**) the relationship of the mean grain size and PLS.

**Figure 14 materials-13-04355-f014:**
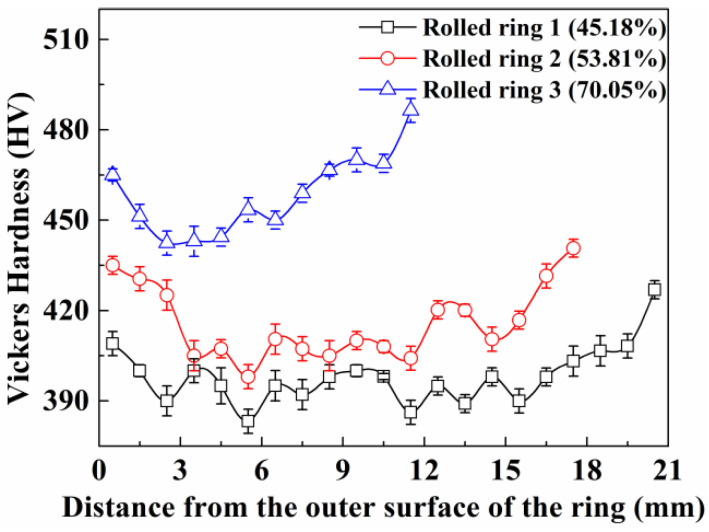
Distribution of the hardness of the rolled rings.

**Figure 15 materials-13-04355-f015:**
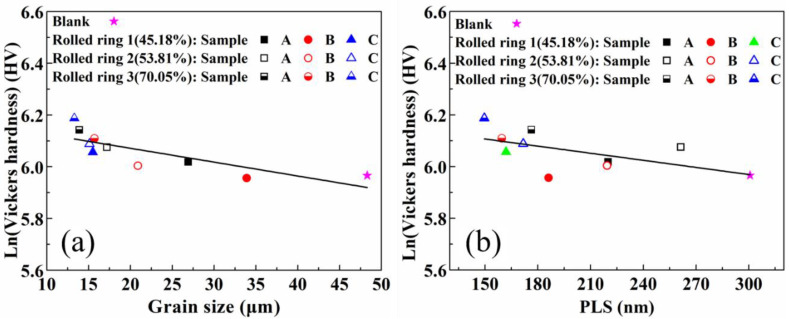
(**a**) Relationship between grain size and Vickers hardness of the samples, (**b**) relationship between PLS and Vickers hardness of the samples.

**Table 1 materials-13-04355-t001:** Processing parameters used in hot ring rolling simulation and experiments.

Rolls and Ring	Parameters	Values and Units
Driven roll	Outer diameter	570 (mm)
	Rotation speed (n)	7.54 (rad/s)
	Feeding speed (v)	3 (mm/s)
Idle roll	Min diameter (D_min_)	50 (mm)
	Max diameter (D_b_)	66 (mm)
	The height of groove ball (H_b_)	32.75 (mm)
Guide roll	Outer diameter	60 (mm)
Ring	Test temperature	1050 (°C)
	Friction coefficient between the rolls and ring	0.4

**Table 2 materials-13-04355-t002:** Dimensions of the rings with different rolling reductions.

Rings	Experimental Value (mm)			Simulative Value (mm)			Error of the Outer Diameter
	D	d_1_	d_2_	D	d_1_	d_2_	
Blank	170.4	91.6	-	170.4	91.6	-	-
Rolled ring 1	217.5	158.3	174.3	215.5	156.9	173.3	0.92%
Rolled ring 2	239.8	187.4	203.4	238.6	186.3	201.1	0.50%
Rolled ring 3	312.9	273.3	289.3	313.6	274.3	290.6	0.22%

**Table 3 materials-13-04355-t003:** The frequency of grain boundaries with different misorientation scales at sample C.

Misorientation	Frequency (%)			
	Blank	Rolled Ring 1 (45.18%)	Rolled Ring 2 (53.81%)	Rolled Ring 3 (70.05%)
<5°	42.7	49.0	40.5	37.9
5–10°	12.7	13.1	12.6	14.8
10–15°	5.7	4.4	4.5	3.3
>15°	38.7	33.3	41.7	43.7

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
