# Peer review of "Influence of Plastic Deformation on Microstructural Evolution of 100Cr6 Bearing Ring in Hot Ring Rolling"

_materials, 2020, doi:10.3390/ma13194355_

Round 1
Reviewer 1 Report
The paper is interesting, but the part with numerical simulation is poorly described. It seems that authors have little knowledge of numerical simulations - it cannot be used as a "black box".
In such form, it is better to remove the numerical simulation completely.Statement that the model is valid is not supported at all. If the model is validated - provide the evidence.The real purpose of the model usage should be explained - paper in fact can be completed without this simulation.
Moreover, the numerical model - if results are presented - should be described, at least.- material model and parameters, temperature dependent (graph of temperature dependent characteristic)- plastic flow law used,- friction in the model.In general - method of calculation used in FEM? Implicit, explicit?So huge plastic probably deformation requires remeshing - should be discussed.If the model is simplified - simplifications have to be explicitly enumerated.
If not properly described - numerical simulation has to be removed from paper - it will increase the value of paper.
It would be interesting to provide more practical conclusions - what should be done to improve the quality of the bearing ring.
Detailed remarks:
Line 72. Why the model is claimed valid? How it was validated? Unsupported, should be explained.
L74: driven roll? Should be pointed on fig 1. Which rolls are idle and driven (probably better driving)?
Fig. 1. arrows with labels needed.
Fig. 2. What is the meaning of colours? Should be explained, with units.
Table 1: dimensions D, d1 and d2 are not defined. Drawing is required. What is the rolling reduction? Equation?
Fig. 8. It has to be clearly stated, that it is a numerical result (in text and description of figure). 6 on the vertical axis means 600% of plastic deformation? Which hypothesis was used to calculate equivalent strain from the strain tensor - von Mises? Tresca? Should be stated in paper.
Author Response
Our answer is in the attachment, please check.

Reviewer 2 Report
Dear Authors!
I have considered your manuscript "Influence of plastic deformation on microstructural evolution of 100Cr6 bearing ring in hot ring rolling " and generally admitting its good quality and soundness I think it is not ready for publication in the present form. I recommend you MAJOR revision addressing my comments given below:
A. Typos and inaccurate phrasing:
Line 54 " and further deformation could not exceed the grain limit. " Intuitively I understand what you want to say, but the sentence is confusing. Please, correct.
Lines 134-139 - This part should be massively shortened. You accurately describe all what is seen in the Figure 7 giving no meaningful information.
Line 163 " the dislocation is improved and provides driving force for dynamic recrystallization " I think you meant dislocation density. The same in the Line 227
Line 193-195 - The phrase absolutely contradicts to the data from the Table 2. I guess it is a technical error.
Line 224. "Expect" -> except
Line 228 "The pearlite colonies and PLS significantly reduce at hot..." -> The size of pearlite colonies and PLS are significantly reduced at hot...
B. MATERIALS AND METHODS section must be seriously revised.
- Lines 84 -86. Please, give more details on the temperatures of upsetting, punching, leveling and rolling. it is obvious that these temperatures are below 1050 C specified. Please, specify the cooling conditions (normalization in air?) after hot rolling.
- Please, draw the positions of "specimens having cylindrical appearance" at the cross-section shown in the Figure 5. What is the diameter of these cylinders?
- EBSD: 1. How many view fields were analyzed? 2. How did you calculate the span of mean grain size in the Figure 7d. 3. Having good EBSD images it is expected you are able to present Pole Figures. Please, provide.
- Line 159 - How the equivalent strain was calculated? Please, provide.
C. RESULTS.
- Line 211 " compared with that of the blank which value is 300.7 nm" - Please, show this PLS in the Figure 4.
- Misorientation angles - How many grains were analyzed? How did you select grains for misorientation angle analysis?
- Table 2 - I am afraid that all the difference in the values of misorientation angles vs reduction is statistically insignificant (and Figure 11 proves this). Therefore, all speculations in the lines 196-204 must be much less definitive but cautious.
- Figure 14 - Please, explain more careful the distance between two adjacent Hardness test points as 1mm. In the Figure 14 the distance between experimental points is obviously much less. How a reader my understand this? What about statistics? How many tests were carried out per a point?
- I strongly recommend you to make and present correlation analysis (in log scale) HV vs grain size and HV vs PLS for your experimental data. This can be the most worth part of your research.
Regards ...
Author Response

(The authors gave the same response as above.)

Reviewer 3 Report
Dear Authors,
This paper doesn't bring any new in scientific terms. It means that novelty doesn't exist and some issues are really old (more than 40 years). The paper seems a didactic report about rolling bearing manufacturing process. The Introduction is the best part of the paper, presenting a scientific framework, but with high didactic content. Some Figures are not needed, because they are very well known and doesn't bring any new to the work.
Figure 4 is in the paper, but it is not referred in the text, and also it is not contextualized. This is just one more justification (among many others) to reject the paper.
Moreover, the paper presents severe lacks of English, being hard to read, with sentences withou sense.
Thus, the paper must be rejected.
Kind regards,
Reviewer
Author Response

(The authors gave the same response as above.)

Round 2
Reviewer 1 Report
If material model is from [11,12] it is OK, however it was for GCr15 (not for 100Cr6 steel), but is is quite similar.
The "straining rates" should be changed to "strain rates"
Description of material model in simulation is still poor, but it can be accepted while the results of simulation are only used to illustrate the change of microstructure.
Author Response

(The authors gave the same response as above.)

Reviewer 2 Report
Dear Authors! I appreciate your intensive work to address my comments in the revised version of your manuscript. I am ready to recognize your paper as ready for publication after MINOR revision. I want you present the results of correlation analysis HV vs grain size and HV vs PLS - did you receive the slope corresponding to theoretical expressions given in the text?
Regards ...
Author Response

(The authors gave the same response as above.)

Reviewer 3 Report
No comments.
Author Response

(The authors gave the same response as above.)
